# Research on the Primary Factors Influencing the Quality of Clinical Coding Under DRG Payment Systems: A Survey Research

**DOI:** 10.3390/healthcare13080849

**Published:** 2025-04-08

**Authors:** Yinghong Fu, Guangying Gao, Huiying Xing, Shanshan Dai, Xinyu Cai, Jiashuai Tian

**Affiliations:** 1School of Public Health, Capital Medical University, No. 10 Xitoutiao, Fengtai District, Beijing 100069, China; fuyinghong@ccmu.edu.cn (Y.F.); daishanshan25@163.com (S.D.); ytlk_serefina927@126.com (X.C.); iridescent.tian77@gmail.com (J.T.); 2Yanjing Medical College, Capital Medical University, No. 4 Dandong Road, Shunyi District, Beijing 101300, China; 17720180306@163.com

**Keywords:** diagnosis-related groups, medical records, clinical coding, influencing factors

## Abstract

**Background:** The main data basis for the Diagnosis-Related Group (DRG) payment methodology is the disease diagnosis and clinical codes in the Medical Record (MR). Problems such as up-coding have arisen during implementation in many countries, and problems with MR quality and solutions have been studied mostly from the physician’s perspective. We investigated and analyzed the main influences on clinical coding from the perspectives of directors and coders in Medical Records Section (MRS) to provide recommendations for improvements in data quality. **Methods:** The questionnaire was developed, revised and improved using literature research and expert consultation methods. From 13 to 19 June 2024, the electronic questionnaire survey was conducted among the directors and coders of medical records department in healthcare organization. A total of 484 directors and coders were included in this study. And relevant statistics were computed and analyzed by non-parametric tests. **Results:** Coders should possess strong job responsibilities (92.36%), coding skills (91.33%), knowledge of clinical medicine (90.70%), and other comprehensive qualities and abilities. When encountering difficult problems, the clinical coders should first communicate with clinical doctors (91.95%). The two main factors affecting the quality of MR and clinical coding are the individual factors of doctors (88.84%) and the individual factors of coders (85.54%). **Conclusions:** Doctors and coders are the primary factors influencing the quality of clinical coding. It is recommended to establish a systematic training program for doctors to enhance the connotative quality of MR, for coders to solidify professional coding skills, strengthen communication and exchange, adopt reasonable behaviors, avoid moral hazards, and effectively improve the quality of clinical coding.

## 1. Introduction

Diagnosis Related Groups (DRG) payment system originated in the United States, refers to the hospitalized patient’s disease according to the diagnosis, gender, age, etc., is divided into a number of groups, and on this basis according to the degree of severity of the disease and its presence or absence of commodities, complications are divided into different levels, each group of different levels are set up for the payment of the corresponding standard, the health insurance agency in accordance with the standard payment to the hospital. The insurance agency pays the hospital according to the standard [1]. Through the unified disease diagnosis classification fixed payment standard, to achieve the standardization of medical resources utilization, prompting hospitals to reduce costs in order to obtain profits, which is conducive to cost control [2].

As reported in the literature, the DRG payment system implemented in some countries suffers from a number of negative problems such as up-coding, cost shifting, and affecting the development of new medical technologies [3,4]. Silverman [5] showed that there was an up-coding behavior among patients enrolled in the Medicare program for the elderly within the healthcare facilities in the U.S.A. after the implementation of DRG. Barros [6] found that there was up-coding behavior in the Portuguese universal health insurance payment to public hospitals using DRG, where the age of the patient was the main influencing factor. Berta [7] evaluated the impact of up-coding behavior on hospital efficiency in Italy. Some undesirable problems in the DRG payment process can be mitigated to some extent by internal hospital audits [8], information technology applications [9], and active participation of clinicians [10], but an in-depth analysis and study of the main influencing factors that generate the quality of clinical coding is still needed.

In China, all the data of DRG are collected from the first page of MR, especially the quality of the primary diagnosis and the clinical coding, is the basis for accurate grouping of DRG [11]. That is the quality of DRG grouping and application depends on the quality of clinical coding. On 26 May 2023, the Notice on Carrying Out Actions to Comprehensively Improve the Quality of Medical Care (2023–2025) (State Healthcare and Medical Affairs (2023) No. 12), the third of the Special Actions: improving the quality of the medical records’ connotation. It is required that, by the end of 2025, the correctness of the coding rate of major diagnoses on the first page of the MR is no less than 90%. At present, significant results have been achieved in exploring the law of DRG payment, implementing the reform tasks and the actual payment, but there are still problems related to the quality of MR, such as defective information [12], hidden up-coding [13] etc., which need to be solved urgently. Guo X. [14], Qin H. [15] and many other scholars have found that coding defects lead to deviations between DRG grouping results and actual diagnostic and treatment behaviors. Huang H. [16] found that interchanging the primary diagnosis with other diagnoses may lead to higher weighting of diseases. Wang T. [17], Hou Q. [18], and Ma Z. [19] analyzed the overall situation of the first page of the MR and found the following quality defects: wrong selection of the main diagnosis, wrong selection of the main surgery or operation, irregular writing, wrong coding of diseases and surgeries. Defective disease diagnosis and surgical operation information directly affects DRG grouping information. Scientific, accurate and timely MR information has become a strong foundation for the effective implementation of the new policies of modern healthcare reform [20,21].

On 26 November 2021, the National Health Security Bureau issued the Notice on Issuing the Three-Year Action Plan for the Reform of DRG/DIP Payment Methods, which explicitly proposed that from 2022 to 2024, the task of reforming the DRG/DIP payment methods would be comprehensively completed, and that the high-quality development of healthcare insurance would be promoted. On 15 March 2022, 66 fixed-point healthcare organizations in Beijing initiated the actual 647 disease group payment. On 9 January 2025, the Notice on Promoting the Reform of Instant Settlement of Basic Medical Insurance Fund (Medical Insurance Office [2025] No. 1), about 80% of the coordinated areas across the country basically realized instant settlement in 2025, and all coordinated areas across the country realized instant settlement in 2026. It is evident that the reform of health insurance payment methods based on MR data is advancing rapidly in China. How to standardize the management of MR information and continuously optimize the quality of the MR has become a key point for the sustained and deepened advancement and success of the reform of the healthcare payment method to pry the changes in the healthcare service system.

Usually, clinicians follow a series of specifications to write medical records and fill in the diagnosis of diseases, surgeries and operations on the first page of MR. On this basis, the coders ensure the accuracy and completeness of the basic data of DRG operation by reading the MR and coding or auditing the codes in strict accordance with the coding rules. Many scholars have studied the problems of MR quality from the perspective of physicians, and found that irregularities in the data on the first page of MR filled in by physicians and the lack of coder competence are the main factors restricting the improvement of the quality of clinical coding. The purpose of this study is to analyze the attitudes and perceptions of coding and the main factors influencing the quality of clinical coding from the perspectives of coders and directors in MRS, and to propose targeted recommendations for improvement, so as to provide a strong guarantee for the smooth operation of the DRG payment system.

## 2. Materials and Methods

### 2.1. Study Design and Participant Recruitment

This cross-sectional survey targeted Medical Records Section (MRS) directors and clinical coders across healthcare institutions in China. Participants were recruited through a national coding professionals’ discussion group facilitated by the Classification of Diseases Group of the Medical Records Committee (CDGMRC). A total of 484 questionnaires were distributed electronically via Questionnaire Star between 13 and 19 June 2024. All questionnaires were returned, yielding a 100% response rate and 100% validity rate after quality control checks.

### 2.2. Data Collection and Measures

#### 2.2.1. Questionnaire Development

The questionnaire was developed through a rigorous three-phase process:

Literature Review: Domestic and international databases were searched using keywords (e.g., clinical coding, medical records, DRG) to extract themes related to MR data quality and coding behaviors, and to initially establish the framework of the questionnaire. Expert Consultation: A panel of 10 specialists validated the questionnaire. 1 DRG policy expert from the National Health Insurance Bureau (NHIB) and Beijing healthcare data quality control. 9 practitioners from tertiary hospitals (8 DRG-implementing institutions, 1 simulation hospital), including 5 MRS directors with coding audit experience and 4 senior coders (>10 years of experience). The panel revised wording, eliminated redundancies, and ensured alignment with DRG payment rules and clinical coding standards. Pilot Testing: The questionnaire underwent two rounds of revisions based on expert feedback and a pre-survey to finalize clarity and relevance.

#### 2.2.2. Questionnaire Structure

The Questionnaire Structure consists of three parts. Basic Information: 12 items (e.g., gender, age, education, coding experience). DRG Knowledge: 3 items assessing understanding of DRG principles, grouping rules, and evaluation metrics (e.g., Case Mix Index). Coding Quality Influencers: 38 items rated on a 5-point Likert scale (1 = strongly disagree, 5 = strongly agree), covering coder perceptions and workflow challenges under DRG reforms.

#### 2.2.3. Data Collection Procedures

The questionnaire was distributed via Questionnaire Star, with quality controls: minimum response time, restrictions on duplicate submissions, and automated screening for invalid responses. Directors of MRS and the CDGMRC facilitated dissemination to ensure serious participation. The questionnaire was anonymous and did not leave any personal information about the respondents.

### 2.3. Data Analysis

Statistical analyses were conducted using SPSS 26.0. Descriptive Statistics includes frequencies and percentages for demographic and Likert-scale data. The reliability and validity of the questionnaire entries were reflected respectively through the Cronbach’s α coefficient and the KMO value. To combine the realities of the research subjects, the reliability and validity of the questionnaire can be usually considered good when the Cronbach’s α coefficient is generally greater than 0.8 [22] and the Kaiser-Meyer-Olkin (KMO) value is generally greater than 0.7. We tested the data for normality and chose either parametric or non-parametric tests based on the normality test results to analyze the primary influencing factors of quality of clinical coding. All the above tests were two-sided, and the significance level was set at 0.05.

### 2.4. Ethical Procedures

This study adhered to ethical standards for human subjects research. The personally identifiable information of all survey respondents was anonymized. The principle of voluntary participation was used to conduct the research study on the basis of informed consent of the survey respondents. Questionnaire Star was able to ensure data security and personal information security during data collection, transmission and storage. The study was conducted in accordance with the Declaration of Helsinki, and approved by the ethics committee of Capital Medical University.

## 3. Results

### 3.1. Basic Characteristics of Respondents

The ratio of male to female survey respondents was about 1:2, mainly concentrated in the age group of 26–35 (35.5%), with undergraduate education in the majority (65.3%), and junior and intermediate titles (70.7%). Professional backgrounds in HIM in the majority (29.5%), followed by clinical medicine (17.6%). The level and type of hospitals where they work are the tertiary general hospitals and public hospitals in the majority. About 3/5 of the coders obtained the coder’s license or training certificate issued by the China Hospital Association (CHA), with 39.6% holding the license for 0–4 years and 26.6% for 5–9 years (see Table 1).

### 3.2. Normality Test

SPSS26.0 software was used to analyze the mean and standard deviation of the data to describe the concentration trend, discrete trend and distribution status of the sample. The results showed that the distribution of the mean value of each measurement item was not balanced, the standard deviation was mostly between 0.7–1.4, and the degree of dispersion of the samples was large. The *p* value of the Kolmogorov-Smirnov test was less than 0.001, indicating that the items did not have the qualities of normal distribution, and the use of the nonparametric test could be considered if it was necessary to compare the variability of the data in different groups.

### 3.3. Awareness of Coding-Related Work

The survey asked respondents about their attitudes and perceptions of clinical coding in the context of DRG implementation in three areas, namely, “The competencies or qualities that clinical coders should possess”, “How to solve difficult problems encountered in the process of coding”, and “How to cope with situations in which a doctor’s diagnosis does not correspond to the coding rules”. The reliability and validity of this part of the questionnaire entries were analyzed, the Cronbach’s α coefficient was 0.893, and the KMO value was 0.934, indicating good reliability and validity.

The mean score for this question was greater than 4, above the level of “Agree”, indicating that the respondents generally agreed with the seven competencies or qualities. 7 options “Strongly Agree” and the sum of percentages of “Strongly Agree” and “Agree” for the seven options was strong sense of job responsibility (92.36%), coding skills (91.33%), knowledge of clinical medicine (90.70%), ability to communicate (88.85%), ability to retrieve information (86.36%), ability to analyze data (84.91%), and knowledge of health statistics (78.93%). Comparison of the respondents with different education levels revealed statistically significant differences in coding skills, knowledge of clinical medicine, communication and expression skills, and a strong sense of job responsibility (*p* < 0.01), while there were no differences in knowledge of health statistics, information retrieval skills, and data analysis skills (*p* > 0.05).

The mean score for each option in the question was greater than 4, indicating general agreement with the six measures to resolve coding challenges. The proportions of “Strongly Agree” and “Agree”, in descending order, were communicating with clinicians (91.95%), discussing within the department (91.32%), consulting a reference book (91.12%), inspecting the coding history (84.92%), searching the Internet (84.91%), consulting an outside expert (84.91%). Comparison of different hospital levels revealed significant differences (*p* < 0.01) in searching for tool books, interdepartmental discussions, communicating with clinicians, and consulting outside experts.

Analysis of coder responses when physicians’ diagnoses does not match coding rules. The highest percentage of “Strongly agree” and “Agree” for the six options in this question item was “communicate with a physician to agree before coding” (see Figure 1).

There were differences in the way coders handled the physician’s diagnosis when encountering a discrepancy between the physician’s diagnosis and the coding rules, depending on their years of coding experience. There was a statistical difference (*p* < 0.01) between different coding work years according to the physician’s diagnosis, coding after communicating with the physician, coding after carefully reading the MR, and entering a higher weight group without considering the coding rules. Comparing different levels of hospitals found that there was a statistically significant difference (*p* < 0.05) in the number of years coding according to the doctor’s diagnosis, coding after communicating with the doctor, coding according to the coding rules tending to enter the higher weight group, and disregarding the coding rules tending to enter the higher weight group. Comparing different age groups found that the difference between coding according to the doctor’s diagnosis, coding after communicating with the doctor in agreement, coding after reading the MR to find the basis of the diagnosis, and disregarding the coding rules into the higher weight group was statistically significant (*p* < 0.01). The results of nonparametric multiple comparisons are shown in Table 2 and Table 3.

### 3.4. Analysis of Influencing Factors of MR and Coding Quality

The “quality of MR” refers to the evaluation standard of the hospital in the process of MR management, to ensure the quality of medical care, service quality, and efficiency by filling out, filing, and inquiring about the MR in a correct, complete, accurate, standardized and reasonable manner. The “quality of clinical coding” refers to the degree of accuracy and standardization of the coding standards followed in the coding process, including the accuracy, standardization, consistency, and precision of the coding. The reliability and validity of this part of the questionnaire entries were analyzed, and the Cronbach’s α coefficient was 0.959 and the KMO value was 0.952, indicating good reliability and validity.

Among the main factors affecting the quality of MR, the sum of the proportion of “Strongly agree” and “Agree” is as follows: individual doctor (88.84%), organizational process (80.99%), hospital management (80.58%), information system and equipment (78.52%), and policy/external factors (77.89%). Comparing different levels of hospitals, we found that doctors’ personal and organizational process factors showed significant differences (*p* < 0.05). The results showed that there were significant differences among individual doctors, hospital management, organizational process and policy/external factors (*p* < 0.01). The results of multiple comparisons are shown in Table 4.

Among the main factors affecting coding quality, the sum of “Strongly agree” and “Agree” is as follows: individual coder (85.54%), organizational process (82.65%), hospital management (80.17%), policy/external (78.71%), information system and equipment (77.27%). Comparing different hospitals, it was found that there were significant differences in the personal factors of coders (*p* < 0.05). The higher the hospital level, the higher the proportion of agreeing that coders were the main factor of coding quality; There was no significant difference in the other 4 items (*p* > 0.05). By comparing different job titles, we found that there are significant differences among individual coders, organizational processes, hospital management, information systems and equipment, and policy/external factors (see Table 5).

The average score of all options is greater than 3 points, indicating that the survey respondents generally agree with the 9 options affecting coders’ accurate coding. The top three “Strongly agree” and “Agree” of the nine options are that the connotation of MR written by doctors is not good, and the coders lack clinical medical knowledge and experience in difficult coding processing (see Figure 2).

Compared with different coding years, we found that there were significant differences in poor connotation quality of MR, poor mastery of coding rules, and lack of clinical knowledge (*p* < 0.05). The longer the coding years, the higher the approval ratio. Comparing different professional titles, we found that the accurate coding of coders was affected by 7 aspects, such as poor connotation quality of MR, poor mastery of coding rules, and insufficient reading of MR content (*p* < 0.01). The results of multiple comparisons are shown in Table 6.

## 4. Discussion

In 2021, National Health Commission of the People’s Republic of China issued the National Healthcare Quality and Safety Improvement Targets, which included the rate of correct coding of the primary diagnosis on the first page of the MR as one of the “Ten Major Targets”. Clinical coders, as translators who translate patient information into alphanumeric codes [23], are a key factor affecting the correct coding rate of primary diagnoses [24]. The basic profile of the coders showed that the highest educational level was dominated by bachelor’s degree accounting for 69.48%, with the highest number of HIM majors, and master’s degree was the next highest accounting for 19.07%, with Clinical Medicine and Preventive Medicine majors dominating. This result is better than the recent relevant literature reports [25,26], indicating that the academic structure and professional background of clinical coders are gradually optimized and improved, but the shortage of highly educated counterparts is still a bottleneck in the development of the clinical coding. Coders under the age of 25 accounted for 24.25% of the total number of coders, junior titles accounted for 41.42% of the total number of coding years and the number of years of coders holding certificates was 0–4 years. It is suggested that at the present stage, when the requirements for the quality of coding have multiplied, it is necessary to strengthen the comprehensive ability training of low seniority coders, and the issue of title promotion should also be given sufficient attention to avoid the problems of staff loss and team instability.

In the context of DRG payment method reform, the comprehensive quality and competence of clinical coders are crucial for improving the quality of clinical coding and promoting the in-depth development of DRG. This survey showed that the top four comprehensive qualities and abilities with the highest percentage were, in order, a strong sense of job responsibility, coding skills, knowledge of clinical medicine, and communication and expression skills, which is consistent with the study by Zhou [27]. Other results found that coders chose to communicate with clinicians the most when they encountered difficult problems, and MR directors considered that inviting experts from outside hospitals to give guidance was also an effective way. Coding skills and clinical medical knowledge, as the most basic skills for coders to carry out coding, constitute the core part of coding professionalism and must be learned continuously. In addition, clinical coders should have a significantly higher sense of job responsibility, strictly abide by the code of ethics and take coding seriously in order to ensure accurate and complete coding results.

When encountering a physician’s diagnosis that is inconsistent with the coding rules, the score for “Tend to go into a higher weight group regardless of the coding rules” was between “Fair” and “Disagree”. “Coding after communicating with the physician” and ‘Reading the chart and finding the basis for the diagnosis before coding’ scored greater than 4, indicating that survey respondents highly agreed with these two actions. “Tend to go into a higher weight group based on coding rules” and “Code according to the physician’s diagnosis” were identified by a significant portion of survey respondents and are worthy of attention. Comparisons revealed that the older the coder and the more years of coding experience, the more likely the coder was to communicate with the physician before coding and to read the chart carefully before coding, whereas the younger the coder and the fewer years of coding experience, the more likely the coder was to code according to the physician’s diagnosis and to enter into a higher weight group without regard to the coding rules. Entry higher weight group is often manifested as up-coding, and this behavior allows to obtain higher payment levels, which is the most serious side effect of DRG payments [28,29,30]. In summary, most coders will use reasonable methods such as communicating with the physician or carefully reviewing the MR to resolve any discrepancies between the physician’s diagnosis and the coding rules. However, we must be paid attention to the possibility that shorter coding years or younger coders may engage in behaviors such as ignoring coding rules and disregarding coding ethics during the coding process [31].

The results showed that the main factors influencing MR quality were individual doctor factors, usually including poor writing of MR and lack of understanding of coding rules etc., with a higher proportion of endorsement in second-level and higher hospitals. The higher the title, the higher the proportion of endorsement for individual doctor, hospital management and organizational process factors. The main factors affecting quality of clinical coding are individual coder factors, which usually include mismatch of professional backgrounds, insufficient communication with physicians etc., and the higher the level of the hospital, the higher the proportion of agreement. Among the factors affecting coders’ accurate coding, the poor quality of physician written chart topped the list, followed by coders’ lack of clinical knowledge and inexperience in handling difficult codes. Directors of MR had a higher percentage of agreement with coders’ failure to master coding rules, insufficient reading of MR, insufficient active communication with physicians, and low motivation, while senior coders had a higher percentage of agreement with poor quality of MR, coders’ lack of proficiency in mastering coding rules, and lack of clinical knowledge.

It can be seen that the internal quality of MR is always an important basis for ensuring the accuracy and completeness of clinical coding. Although the cognitive level of physicians and the internal quality of MR have been improved to some extent since the implementation of the DRG [32], they are still important connotations that must be emphasized and continuously focused on and upgraded.

The cultivation of professionals related to clinical coding needs to combine multidisciplinary backgrounds such as medicine, informatics, management, and computer science [33]. For example, in the United States, clinical coders are usually required to have a bachelor’s degree in Medicine, HIM, Nursing, Biomedical Sciences, and other majors, with higher educational requirements for clinical coders [34]. Break the traditional single promotion mode, establish the mutual conversion mechanism among management positions, professional and technical positions, and work skill positions, and provide more promotion options for clinical coders. By strengthening the quality control of the first page of the case, coding training, and other ways to prevent the moral risk of up-coding, to ensure the benign operation of the DRG payment system [35,36]. Physicians and coders are the two core subjects of MR and an important breakthrough for improving the quality of MR. Strengthen the supervision and management of clinical coding, incorporate the quality of clinical coding into the performance appraisal system of clinicians and coders, commend and reward clinicians and coders with excellent performance, and criticize, educate, and punish individuals with problems.

The results of this study can provide theoretical and practical support from various aspects. Hospital administrators can refer to optimize the quality control process of in-hospital coding, clinicians can improve the standardization of medical record writing, coders can enhance their clinical knowledge reserve and communication skills, and health insurance regulators can dynamically optimize the DRG grouping rules and improve the audit mechanism.

This study analyzed the main influencing factors of coding quality mainly from the perspectives of coders and MR managers, which provides a reference for the improvement of coding quality, but there are still some limitations in the selection of survey subjects. This study analyzes the main factors affecting the quality of clinical coding only from the perspectives of the director of MR and coders. It did not consider the impact of the problems of DRG itself on coding quality in the implementation of health insurance payment reform. The problems of DRG itself include: unclear grouping rules, poor feedback channels for grouping problems, and impact on the objectivity of coding, etc. These problems directly or indirectly affect the management behaviors of the hospitals, the medical behaviors of the doctors, and the behaviors of the coders, which in turn affect the quality of the coding. Additionally, the respondents of this study mainly focused on tertiary hospitals, and the sample coverage was relatively narrow. Future studies could further expand the sample to cover more hospitals of different levels and types to enhance the generalizability and representativeness of the findings.

## 5. Conclusions

In the context of DRG payment, the quality of clinical coding is crucial, and the role of coders is becoming more and more prominent. Clinical coders should possess coding skills, knowledge of clinical medicine, communication and presentation skills, and a strong sense of job responsibility. Coders can solve problems encountered in the coding process by searching for tools, discussing within the department, communicating with clinicians, and consulting with outside experts. However, in the process of coding, coders must follow the coding rules, read the medical records carefully to find the diagnostic basis or communicate with the doctors before coding. Most coders will adopt a reasonable approach to solve difficult problems in their work, but the overall status of coders is not optimistic, and the possible irrational behaviors of low-seniority coders should also be actively paid attention to. At the same time, constructing a synergistic mechanism between doctors and coders is conducive to improving the data quality of clinical coding.

## Figures and Tables

**Figure 1 healthcare-13-00849-f001:**
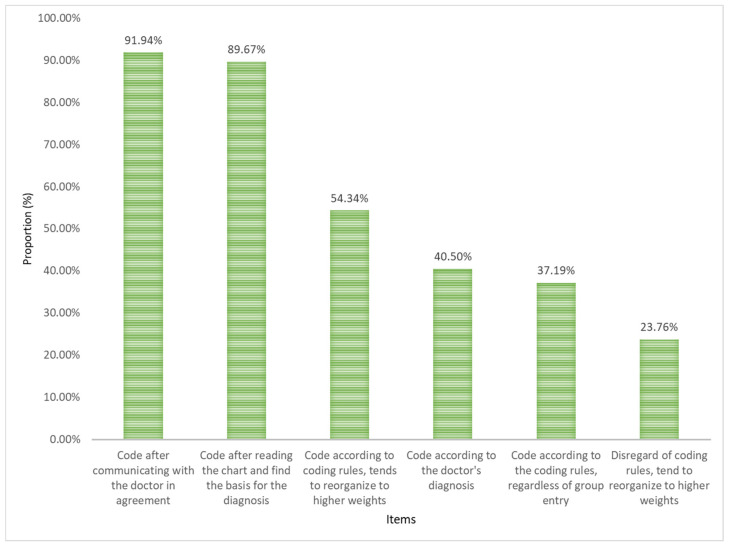
The proportion of respondents who agreed with the solution that the doctor’s diagnosis did not match the coding rules.

**Figure 2 healthcare-13-00849-f002:**
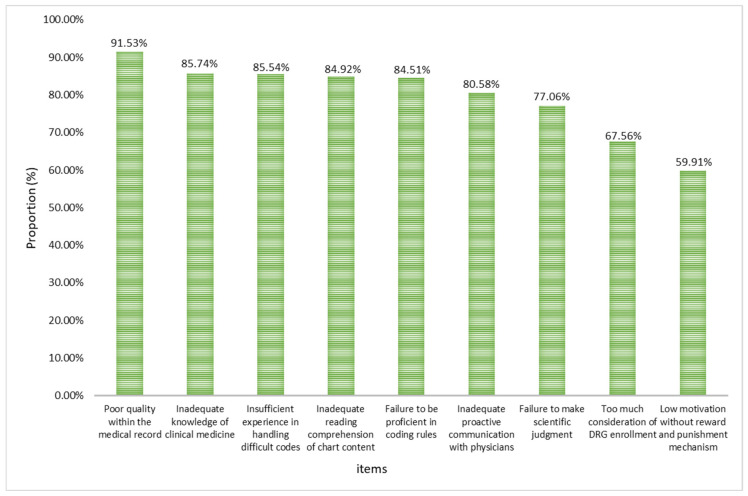
The proportion of main factors affecting coders’ accurate coding.

**Table 1 healthcare-13-00849-t001:** Basic information of respondents (*n* = 484).

Items		Number of People	Proportion (%)
Sex	Male	163	33.7
	Female	321	66.3
Age group	≤25	111	22.9
	26~35	172	35.5
	36~45	121	25.0
	46~55	69	14.3
	≥56	11	2.3
Education level	Post-secondary and below	10	2.1
	Junior college education	41	8.5
	Bachelor Degree	316	65.3
	postgraduate	105	21.7
	PhD student	12	2.5
Title	Senior	22	4.5
	Deputy Senior	53	11.0
	Intermediate	168	34.7
	Junior	174	36.0
	None	67	13.8
Years of coding *	≤4	172	46.9
(*n* = 367)	5–9	89	24.3
	10–14	53	14.4
	15–19	31	8.4
	≥20	22	6.0
Hospital level	Tertiary hospital	375	77.5
	Secondary Hospital	68	14.0
	First level hospital	14	2.9
	Unrated hospital	27	5.6

* Only for coders to fill in.

**Table 2 healthcare-13-00849-t002:** Coders with different coding years deal with non-parametric multiple comparisons between a doctor’s diagnosis and coding rules.

Items	Years of Coding Rank Mean					Kruskal-Wallis Test Statistic *h*-Value	*p*-Value
	≤4 (*n* = 172)	5–9 (*n* = 90)	10–14 (*n* = 53)	15–19 (*n* = 31)	≥20 (*n* = 22)		
Code according to the doctor’s diagnosis	210.92 ^a^	176.89	148.97 ^b^	154.47 ^c^	137.00	25.362	0.000 **
Code after communicating with the doctor in agreement	155.72 ^a^	211.21 ^b^	222.65 ^c^	203.94	180.91	34.804	0.000 **
Code after reading the chart and find the basis for the diagnosis	163.63 ^a^	207.63 ^b^	197.70	192.27	210.32	16.661	0.002 **
Code according to the coding rules, regardless of group entry	197.63	178.41	178.95	155.68	160.75	6.797	0.147
Code according to coding rules, tends to reorganize to higher weights	186.37	182.43	178.54	187.81	188.02	0.330	0.988
Disregard of coding rules, tend to reorganize to higher weights	221.01 ^a^	154.71 ^b^	150.42 ^b^	154.55 ^b^	145.23 ^b^	40.542	0.000 **

** *p* < 0.01, statistically significant differences between a and b comparisons and a and c comparisons in the same program.

**Table 3 healthcare-13-00849-t003:** Coders of different ages respond to non-parametric multiple comparisons between doctor’s diagnosis and coding rules.

Items	Age Rank Mean					Kruskal-Wallis Test Statistic *h*-Value	*p*-Value
	≤25 (*n* = 111)	26–35 (*n* = 172)	36–45 (*n* = 121)	46–55 (*n* = 69)	≥56 (*n* = 11)		
Code according to the doctor’s diagnosis	306.18 ^a^	257.15 ^d^	206.14 ^b^	177.12 ^c^	181.00	53.181	0.000 **
Code after communicating with the doctor in agreement	176.27 ^a^	242.24 ^b^	287.41 ^b^	259.02 ^b^	317.27 ^b^	55.658	0.000 **
Code after reading the chart and find the basis for the diagnosis	190.93 ^a^	244.73 ^b^	272.12 ^b^	262.83 ^b^	274.59	28.390	0.000 **
Code according to the coding rules, regardless of group entry	272.12	243.58	227.36	221.62	224.32	8.576	0.073
Code according to coding rules, tends to reorganize to higher weights	255.63	240.46	238.16	230.72	263.50	1.997	0.736
Disregard of coding rules, tend to reorganize to higher weights	314.75 ^a^	248.66 ^b^	209.24 ^c^	174.43 ^e^	210.05	57.037	0.000 **

** *p* < 0.01, statistically significant differences between a and b, a and c, a and e, b and d, c and d, b and e in the same program.

**Table 4 healthcare-13-00849-t004:** Non-parametric multiple comparisons of factors affecting the quality of MR considered by respondents of different professional titles.

Influencing Factors	Title Rank Mean					Kruskal-Wallis Test Statistic *h*-Value	*p*-Value
	Senior (*n* = 22)	Deputy Senior (*n* = 53)	Intermediate (*n* = 168)	Junior (*n* = 174)	None (*n* = 67)		
Individual physician	223.07	293.24 ^a^	259.45 ^a^	237.75 ^a^	178.57 ^b^	29.551	0.000 **
Information system and equipment	247.95	262.56	252.12	240.57	205.74	7.554	0.109
Hospital management	232.11	313.10 ^a^	261.50 ^c^	220.29 ^b^	200.10 ^d^	31.516	0.000 **
Organizational processes	234.14	307.01 ^a^	255.12 ^c^	229.38 ^b^	196.65 ^d^	24.838	0.000 **
Policy/External factors	243.82	282.15	261.41	224.51	209.99	15.833	0.003 **

** *p* < 0.01, statistically significant differences between a and b, a and d, c and d in the same program.

**Table 5 healthcare-13-00849-t005:** Non-parametric multiple comparisons of factors affecting coding quality among respondents with different professional titles.

Influencing Factors	Title Rank Mean					Kruskal-Wallis Test Statistic *h*-Value	*p*-Value
	Senior (*n* = 22)	Deputy Senior (*n* = 53)	Intermediate (*n* = 168)	Junior (*n* = 174)	None (*n* = 67)		
Individual coder	279.91	312.31 ^a^	258.94 ^d^	222.14 ^c^	186.81 ^b^	37.341	0.000 **
Information systems and equipment	261.61	278.58 ^a^	249.07	237.55	204.07 ^b^	10.982	0.027 *
Hospital management	237.66	290.59 ^a^	252.14	231.58	210.25 ^b^	13.587	0.009 **
Organizational process	243.45	287.10 ^a^	258.23	228.65	203.43 ^b^	17.005	0.002 **
Policy/External Factors	275.27	272.29	255.87	229.95	207.25	12.504	0.014 *

** *p* < 0.01, * *p* < 0.05, statistically significant differences between a and b, a and c, b and d in the same program.

**Table 6 healthcare-13-00849-t006:** Non-parametric multiple comparisons of the main influencing factors of precision coding considered by respondents of different professional titles.

Influencing Factors	Title Rank Mean					Kruskal-Wallis Test Statistic *h*-Value	*p*-Value
	Senior (*n* = 22)	Deputy Senior (*n* = 53)	Intermediate (*n* = 168)	Junior (*n* = 174)	None (*n* = 67)		
Poor quality within the MR	252.20	296.66 ^a^	264.12 ^d^	234.30 ^b^	163.57 ^c^	43.163	0.000 **
Not proficient in coding rules	280.64	307.81 ^a^	254.13	223.68 ^b^	198.03 ^b^	28.771	0.000 **
Inadequate comprehension of chart	270.89	308.40 ^a^	256.56	219.83 ^b^	204.68 ^b^	28.204	0.000 **
Lacking experience with difficult codes	267.20	301.04 ^a^	255.57	225.47 ^b^	199.52 ^b^	24.171	0.000 **
Lacking knowledge of clinical medical	269.68	297.42 ^a^	260.73	218.29 ^b^	207.29 ^b^	25.353	0.000 **
Lacking communication with doctors	280.09	317.47 ^a^	253.83 ^b^	218.36 ^b^	205.15 ^b^	32.174	0.000 **
Not make a scientific judgment	270.55	309.59 ^a^	246.32	226.12 ^b^	213.19 ^b^	21.095	0.000 **
Lack of incentives and penalties	302.68	269.60	238.79	227.21	250.32	9.124	0.058
Overthinking DRG enrollment	247.27	256.44	243.96	241.27	229.42	1.291	0.863

** *p* < 0.01, statistically significant differences between a and b, a and c, b and d, c and d in the same program.

## Data Availability

The dataset is available from the authors upon request. The raw data supporting the conclusions of this article will be made available by the authors upon request.

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
