# Peer review of "Research on the Primary Factors Influencing the Quality of Clinical Coding Under DRG Payment Systems: A Survey Research"

_healthcare, 2025, doi:10.3390/healthcare13080849_

Round 1
Reviewer 1 Report
Comments and Suggestions for Authors
Firstly, allow me to express my gratitude for the opportunity to contribute to the review of this manuscript. I would also like to commend the authors for their work, particularly for addressing such a pertinent topic. Issues surrounding the payment and cost determination of healthcare services, whether in private systems (involving insurance providers) or publicly funded systems, are of significant importance. Understanding and deepening our knowledge of these dynamics is crucial for enhancing efficiency and achieving a fair balance between the quality of care provided and its financial value.
I recommend a minor revision of the Abstract. In my view, demographic data within the results section is not the most relevant information for an abstract. Readers primarily seek to understand whether the manuscript presents significant evidence related to the topic or research question.
The Materials and Methods section requires considerable restructuring. This suggestion does not critique the authors' methodological choices but rather aims to improve the organisation of the information regarding the adopted procedures. Currently, the section appears somewhat unclear.
Regarding the Introduction, I have no comments. It is concise yet comprehensive, effectively framing the topic and its relevance. To enhance clarity and coherence, I suggest reorganising the section according to the following structure:
- Study Design and Participant Recruitment
- Data Collection and Measures (within this subsection, I recommend introducing an additional level titled Questionnaire Development, where all procedures related to the questionnaire’s creation are consolidated and described, improving clarity and understanding)
- Data Analysis
- Ethical Procedures
The Results section is well-written and logically structured, supported by clear and easy-to-read tables. Similarly, the Discussion and Conclusion sections are coherent and well-developed.
However, I strongly recommend including a subsection within the Discussion that explicitly addresses the limitations of the study, as these are inevitably present. Additionally, either within the Discussion or the Conclusion section, it would be valuable to elaborate on the implications of this study for future research.
Overall, in my opinion, with these structural improvements, the manuscript will achieve greater clarity and coherence, strengthening its contribution to the field.
Author Response
Comments 1: I recommend a minor revision of the Abstract. In my view, demographic data within the results section is not the most relevant information for an abstract. Readers primarily seek to understand whether the manuscript presents significant evidence related to the topic or research question.
Response 1: Thank you for pointing this out. We agree with this comment. Therefore, we have removing demographic data within the results section and retain only the most relevant information of the Abstract.
Comments 2: The Materials and Methods section requires considerable restructuring. This suggestion does not critique the authors' methodological choices but rather aims to improve the organization of the information regarding the adopted procedures. Currently, the section appears somewhat unclear.
Response 2: Thank you for pointing this out. We agree with this comment. Therefore, we have restructured the Materials and Methods section. The content has been added and improved in 108-161 lines and has been marked in red.
Comments 3: Regarding the Introduction, I have no comments. It is concise yet comprehensive, effectively framing the topic and its relevance. To enhance clarity and coherence, I suggest reorganizing the section.
Response 3: Regarding the Introduction, I have done a more detailed and comprehensive compilation and elaboration, and added the arguments why it is necessary to conduct relevant research in China in the context of international DRG payments. However, I think you want me to reorganize the “Materials and Methods” section, and I am very much in favor of this and have completed the content to better reflect the materials and research methods of this study. If I have not understood your comments correctly, I look forward to hearing from you again about them.
Comments 4: I strongly recommend including a subsection within the Discussion that explicitly addresses the limitations of the study, as these are inevitably present. Additionally, either within the Discussion or the Conclusion section, it would be valuable to elaborate on the implications of this study for future research.
Response 4: We strongly agree that the limitations of this study should be clearly stated in the Discussion section, and we did mention the limitations, but they were not clear enough. The content has been added and improved in 368-374 lines and has been marked in red. In addition, the value and implications of this study need to be clearly stated. The content has been added and improved in 377-382 lines and has been marked in red.
Reviewer 2 Report
Comments and Suggestions for Authors
Introduction: It would be interesting to go into the fundamental problems of DRG billing in more detail in the introductory chapter and also to point out other types of billing systems with their advantages and disadvantages.
Results, Table 1: Please try to format the numerical values ​​in the right column (proportion) a little better.
Discussion: The reviewer considers the validity of the study to be rather limited, especially since the type of analysis does indeed assess the quality of the assignment of the person carrying out the DRG, as well as some of their characteristics such as gender, age and level of education, but the value of the DRG itself is less assessed or questioned. This fact should be explicitly mentioned in the discussion chapter.
Author Response
Comments 1: Results, Table 1: Please try to format the numerical values in the right column (proportion) a little better.
Response 1: Thank you for pointing this out. We have reformatted and optimized all tables.
Comments 2: Discussion: The reviewer considers the validity of the study to be rather limited, especially since the type of analysis does indeed assess the quality of the assignment of the person carrying out the DRG, as well as some of their characteristics such as gender, age and level of education, but the value of the DRG itself is less assessed or questioned. This fact should be explicitly mentioned in the discussion chapter.
Response 2: Thank you for pointing this out. We have analyzed and evaluated the value and problems of DRG itself and identified it’s direct or indirect impact on coding quality. The content has been added in 387-393 lines and has been marked in red.
Reviewer 3 Report
Comments and Suggestions for Authors
Your findings are in line with studies done previously related to MRs, the poor quality of diagnosis provided by physicians, and the importance of working very closely and communicating with physicians before coding.
I would have liked to see the answers to the three questions you posed earlier as part of your study below:
"The competencies or qualities that clinical coders should possess",
"How to solve difficult problems encountered in the process of coding", and
"How to cope with situations in which a doctor's diagnosis does not correspond to the coding rules".
Since you posed these questions, attempt to provide a response based on your findings in the conclusion.
Comments on the Quality of English LanguageThe English is a bit rough and some of the sentences are hard to understand.
Author Response
Comments 1: I would have liked to see the answers to the three questions you posed earlier as part of your study.
Response 1: Thank you for pointing this out. We have provided clear answers to these three key questions in the concluding section. The content has been added in 399-405 lines and has been marked in red.
Reviewer 4 Report
Comments and Suggestions for Authors
Dear Author(s),
Please, find below some suggestions of improvement of your manuscript:
- line 20- I think is organizations
- line 90- irrelevant to put the response rate if it was 100%
- line 142- Cronbach alpha has an acceptable threshold of 0.70, but maybe the author(s) can provide some references for the threshold of 0.80.
- why did the author(s) use the nonparametric tests? Have you checked the normality of the quantitative variables? If yes, how? Otherwise, there is no sense in using nonparametric tests.
- the Discussion section should be improved with 2 other subsections: limitations of the study and further research. For the limitations, I would include the limited generalizability of the results and the limitations of the cross-sectional study. The further research section should answer the questions: Who and how can the results of the study be used? In what context? What is lacking now?
- In the Introduction section it would be reccommendable to place the research in the context, respectively in China and to bring more arguments why is the research required in China?
Thank you and good luck!
Author Response
Comments 1: line 20- I think is organizations. line 90- irrelevant to put the response rate if it was 100%.
Response 1: Thank you for pointing this out. We've changed the content of these two lines and has been marked in red.
Comments 2: line 142-Cronbach alpha has an acceptable threshold of 0.70, but maybe the author(s) can provide some references for the threshold of 0.80.
Response 2: Thank you for pointing this out. We've provided some references for the threshold of 0.80 and the line 149 has been marked in red.
Comments 3: Why did the author(s) use the non-parametric tests? Have you checked the normality of the quantitative variables? If yes, how? Otherwise, there is no sense in using non-parametric tests.
Response 3: Thank you for pointing this out. We performed a normality test on the data before the non-parametric test, but it was not reflected in the manuscript. Based on your suggestion, we added the normality test results and the line 174-182 has been marked in red.
Comments 4: the Discussion section should be improved with 2 other subsections: limitations of the study and further research. For the limitations, I would include the limited generalizability of the results and the limitations of the cross-sectional study. The further research section should answer the questions: Who and how can the results of the study be used? In what context? What is lacking now?
Response 4: Thank you for pointing this out. We combed through the limitations of the study in detail and the line 383-396 has been marked in red. At the same time, we further clarified the questions that the research component should answer and the line 377-382 has been marked in red.
Comments 5: In the Introduction section it would be rec-commendable to place the research in the context, respectively in China and to bring more arguments why is the research required in China?
Response 5: Thank you for pointing this out. We have organized the introductory section to make clear the importance and necessity of conducting the present study in China and the line 58-91 has been marked in red.
Round 2
Reviewer 4 Report
Comments and Suggestions for Authors
All comments have been properly addressed.